# The Role of DRT in European Urban Public Transport Systems—A Comparison between Tampere, Braunschweig and Genoa

Tiziano Pavanini [1,*], Heikki Liimatainen [2], Nina Sievers [3] and Jan Peter Heemsoth [3]

1 Italian Center of Excellence on Logistics, Transport and Infrastructures (C.I.E.L.I), University of Genoa, 16126 Genoa, Italy
2 Transport Research Centre Verne, Tampere University, 33014 Tampere, Finland; heikki.liimatainen@tuni.fi
3 Institut für Verkehrswesen, Eisenbahnbau und-Betrieb, Technische Universität Carolo-Wilhelmina zu Braunschweig, 38106 Braunschweig, Germany; nina.sievers@tu-braunschweig.de (N.S.); j.heemsoth@tu-braunschweig.de (J.P.H.)
* Correspondence: tiziano.pavanini@edu.unige.it

**Abstract:** Demand-Responsive Transport (DRT) is one of the most valid solutions to tackle the problems affecting public transport today, both in urban and rural areas. Despite its undoubted advantages, it still remains underdeveloped compared to its great potential. The purpose of this paper is to understand the role that on-demand transport plays in the strategic choices of public transport authorities (PTAs): to this end, this study examined the DRT services of three geographically distant European cities, in order to test different social, cultural and regulatory backgrounds, examining their main characteristics. Tampere, Braunschweig and Genoa were selected for the purposes of this work; data and information were collected by viewing the official websites of public transport companies and by direct contact via mail/telephone with the managers responsible for on-call transport. The data collected were then analyzed based on specific Key Performance Indicators (KPIs) identified in academic literature. The results of this paper show that the role of on-call service in the strategic decisions of PTAs depends on the cultural context of reference; some cities focus more on urban services, others on rural transport. In all three case studies examined, on-demand transport is an important aspect of local mobility and with wide room for growth.

**Keywords:** demand responsive transport; on-demand services; public transport; dial-a-ride; urban mobility

## 1. Introduction

Demand Responsive Transport (DRT) has for years been one of the viable solutions to some of the problems that affect mobility in cities and rural areas. DRT allows public transport authorities (PTAs) to optimize the use of vehicles and provide services in areas or to customers that are not covered by scheduled public transport routes. A DRT system well integrated with traditional service discourages the use of private vehicles with consequent air pollution and land-use reduction.

The application of DRT technology within urban and extra-urban public transport is treated in the literature under many aspects (technological, user experience, drivers and barriers, etc.), but the strategic nature of DRT within urban public transport systems has not been widely studied. This international work aims to answer the following research question:

*RQ: Is DRT considered by PTAs as a strategic development area for the future or as an auxiliary service not worthy of investments?*

The present paper tries to address this topic through the application of selected Key Performance Indicators (KPIs), identified in the literature, to the PTAs of three cities

geographically located in Northern Europe, Central Europe and Southern Europe: Tampere, Braunschweig and Genoa, respectively. The data were collected through the analysis of the official websites of PTAs and direct contact via email or telephone with the managers responsible for on-call transport in each of city observed.

The study's originality stems from the selection and investigation of three European cities belonging to different geographical contexts and spheres of cultural influence: in fact, the research work allowed studying the weight that DRT technology covers within the strategic managerial decisions of three LPT companies.

This investigation thus enables comprehending the state of the art of DRT service in the European context, despite the aforementioned cultural and spatial differences that might alter some dynamics.

This paper consists of five sections. Section 1 highlights the need for a deeper understanding of DRT technology as a solution to some issues affecting urban public transport nowadays. Section 2, "Background and Literature Review", presents a brief review of DRT academic literature and the recent developments of this technology in Europe, stating the best cases and providing an interpretation of where this field is headed. Furthermore, this section analyses the typologies of DRT and service models identified by scholars. Section 3, "Materials and methods", presents the main key performance indicators used in the literature to assess DRT performances and introduces KPIs selected for this study. Section 4 illustrates the current configuration of DRT service in the three European cities examined, explaining its characteristics, typology and service model. Furthermore, the data collected for this study are presented: in addition, a comparison between three examined cities is offered to understand the different degrees of priority that DRT plays in PTAs' strategies. Section 5 discusses the results obtained and presents conclusions of this work, including limitations and research agendas.

## 2. Background and Literature Review

Demand-Responsive Transport, also known globally as dial-a-ride [1] or paratransit [2], is a sustainable mobility tool that can be implemented by PTAs to support and integrate traditional scheduled public transport routes with a system where stops and timing of the service are set around the requests of passengers [3].

DRT technology, although introduced already in the 1960s to meet the transport needs of particular categories of users such as the elderly and the disabled, and before that in 1916 at a complete prototype level [4], finds increasing application with the evolution of the IT sector. In fact, it is in the past few decades that DRT has found increased application in both urban and extra-urban contexts and for different purposes.

DRT service generally is carried out with small and eco-sustainable vehicles, combining the economic convenience of traditional buses with the flexibility of taxis: it represents a compromise between these two modalities [5]. DRT aims to achieve mainly two objectives: first, to minimize the operating costs, which increase or decrease according to the flexibility of the service, and subsequently to maximize the quality of the service offered to users, which decreases if waiting or travel times lengthen [6].

Several studies in the literature focus on passengers' travel behaviour, trying to outline target users potentially interested in using DRT service [7], while others apply statistical and mathematical models to predict DRT transport demand: areas with a high density of employment and education facilities, and with a high percentage of car ownership, are more suitable for the deployment of on-demand services. This could be explained by the fact that these areas have a higher need for flexible and personalized transportation solutions, especially during peak hours when demand for public transport is high. In contrast, areas with a high concentration of residential properties or low employment density may require a more traditional fixed-route service, as the demand for personalized on-demand transport may not be as high.

Overall, the susceptibility analysis developed by [8] provides a useful tool to better understand the demand patterns of DRT services in urban areas, and to identify the most

suitable locations for their implementation. This can help transportation planners and operators to optimize their service offerings, reduce costs, and enhance the quality and accessibility of public transport services, as happened in the city of Melbourne.

Furthermore, the case of the North Bristol industrial area, analyzed by [9], highlights the importance of using mathematical and statistical models to tackle complex mobility challenges in urban areas. By combining data from different sources and using advanced modelling techniques, it is possible to gain a better understanding of the mobility needs of different groups of users, identify potential barriers to access and design more efficient and sustainable mobility solutions. This approach can also help to promote more seamless and integrated transport systems, making it easier for people to travel around cities and reducing the negative environmental and social impacts of transportation.

The ABM (Agent-Based Model) model developed by the authors also allowed for the evaluation of different scenarios, such as variations in the number of vehicles available for the on-call service or the impact of the introduction of this system on the traditional public transport network. The model proved to be a very useful tool for decision making, providing valuable insights into the potential benefits, costs and trade-offs associated with the introduction of new mobility solutions.

Another predicting model was proposed by [10] in South Sweden: it showed that it can serve as a useful tool for public transport planners and policymakers in evaluating the potential demand for DRT services in rural areas. The authors recommended the model to be used in conjunction with other data sources, such as demographic and socio-economic data, to provide a comprehensive understanding of the transport needs of rural communities.

Comparing the results of the optimized last/first-mile service with a traditional bus service, the former performs better in terms of waiting time and travel time for passengers. The analysis conducted in Sicily by [11], highlighted the importance of considering the dynamic nature of transportation demand when designing last/first-mile services, as well as the potential benefits of using agent-based models and optimization algorithms in transportation planning. Overall, the findings of this study contribute to the development of more efficient and effective urban transportation systems: the authors created two distinct routes, each with a 30-min travel time, to close the gap in the fixed transit system.

Some authors provide PTA management with valuable assessment frameworks to facilitate investment decisions in DRT service, in an effort to emphasize the critical elements influencing a DRT service's success or failure.

How a PTA evaluates the adoption of a DRT service to augment or replace FT was the major topic that [12] sought to address in their work. As a result, the authors created a framework consisting of four questions that public transportation authorities must answer in order to deeply understand the feasibility of DRT service in their context ("Which is/are the area(s) DRT investments should be examined for? Which is the type and network of DRT investment required for the network? Which is the appropriate evaluation method to be used for the assessment? How should 'success' be defined for deciding upon the implementation of the DRT service?"). Each question may be resolved by obtaining various inputs and data.

According to [13], who conducted research for DRT deployment in England, those who are disabled, commuters, and residents of sparsely populated regions are the groups of individuals most likely to switch their method of transportation and utilize DRT technology. Furthermore, the results of the authors' investigation, which came from the application of an ordered logit model to a regional survey, revealed that men tend to utilize DRT more in retirement than during their working years.

Furthermore, some papers compare the characteristics and performances of Fixed Transport (FT) and DRT service: the study conducted by [4] showed that the majority of users were satisfied with the DRT service experimented in two districts of Amsterdam, particularly in terms of flexibility and convenience. The service was particularly popular

among older adults and people with reduced mobility, who found it easier to access the service and to travel more comfortably than with traditional public transport.

The study also highlighted the potential of DRT in improving mobility in suburban and rural areas, where traditional public transport systems are often insufficient, infrequent or non-existent. By providing a flexible, demand-responsive service, DRT can address some of the challenges of accessing essential services and employment opportunities, particularly for disadvantaged and vulnerable groups.

Overall, the results of the Amsterdam trial demonstrate that DRT has the potential to complement or even replace traditional public transport systems, particularly in areas with low demand or poor connectivity. The service can lead to a reduction in costs, emissions and travel time while offering users greater convenience and comfort. Therefore, the integration of DRT into urban mobility plans and transport policies is an important step towards creating more sustainable, accessible and user-centred public transport systems.

Furthermore, the study conducted by [14] highlights the importance of evaluating the performance of DRT as a stand-alone service, separate from traditional fixed transport modes. By providing a specific assessment framework for DRT, PTAs are now able to make more informed decisions about the implementation of these services. The case study of Breng Flex in the Arnhem-Nijmegen Region demonstrates the potential benefits of DRT, particularly in reducing travel times. This analysis can help guide future investments in DRT, determining whether it can effectively complement or replace existing modes of transportation.

In conclusion, the outbreak of the COVID-19 pandemic, which required the authorities to rethink the current public transport system and reduce the use of private cars, has been an important driver in the diffusion of DRT transport globally [15]. For example, in the English town of Milton Keynes, the pandemic was an opportunity to innovate local public transport [16]: the fixed transport service serving a specific area of the city was converted into a DRT service through the collaboration between the private DRT operator Via and the city council. The success of the initiative led to the confirmation of the pilot and the extension of the lines served by DRT.

Additionally, the potential of the DRT service in the pandemic era has also been studied by [17]: the authors, after conducting a literature review of the past 20 years at the European level to describe the main features of the service, conclude that the most important barriers to DRT development are both normative and socio-economic. The authors also suggest, in order to spread the use of this technology, to try to attract new categories of users, to use innovative ICT systems applied to transport and to evaluate the economic convenience of introducing alternative forms of mobility to FT.

## 2.1. DRT Development in Case Countries

The evolution of DRT is strongly linked to technological development in communication and information technology. The first attempts in Europe, based on American experiments, to integrate on-demand transport into the traditional system date back to the 1970s [18], but it was only in the following decade that DRT began, thanks to developments in the IT field, to take the form of the service it presents today [4]. The first pilot project of this type registered in Italy dates back to 1987, with the introduction in Val Nure (Province of Piacenza) of an additional service to scheduled transport in weak-demand areas [19].

With the evolution of the aforementioned technologies, DRT service found increasing application during the 1990s in all European countries to cope with the inefficient connections of scheduled transport both in urban and extra-urban areas. The pilot projects involved Finland (towns of Seinajoki and Tuusula-Kerva-Jarvenpaa) and Italy (Florence and Campo Bisenzio) [3,20]. The results of these initiatives showed that users are open to change and to test alternative modes of transport and that a DRT service with a low-ticket price, although it can carry fewer passengers than FT, can also be economically viable [20].

Due to the global financial crisis of 2008, many transport companies have suffered heavy cuts in government subsidies and, taking advantage of the simultaneous mass

diffusion of smartphones among the population, have found in DRT service a solution to achieve lower operating costs and a higher-performing load factor of vehicles [21]. DRT trials carried out in the past decade are various. The DRT service "ProntoBus" was tested in the three-year period of 2010–2013 in the municipality of Perugia (Italy). During this pilot, the number of passengers/km doubled, and the cost/km decreased, allowing the service to become permanent even after the end of the experimentation [22,23]. In Finland, Kutsuplus, a pilot project managed by the PTA of Helsinki (HSL) and defined as "apparently the world's first . . . real-time demand-responsive public transport service" [24], started in 2013. The service allowed users to book trips up to 30 min before departure. Due to the limited budget available to local municipalities, this DRT trial was discontinued two years after it started [24].

König and Grippenkoven studied the evaluation of customers' travel behaviour and how it changes with the introduction of DRT [7]. The authors stated that despite the advantages of this technology, it remains little used because, in their opinion, users' points of view and psychological motivations as bases of travel choices have not been widely analysed. They administered a questionnaire to 205 users in two German rural areas and analysed the data. This work concluded that the greatest impact on users' travel choices is "Performance Expectancy", i.e., how much DRT service contributes to improving users' current transport situation.

*2.2. DRT Service Models*

The fields of application of DRT technology were divided into four different typologies [25]:

- Interchange DRT: A feeder service that provides connections to traditional transport (e.g., a shuttle connecting the city centre with the airport or the main train station);
- Network DRT: An additional service or replacement of traditional services considered inefficient from an economic point of view in particular residential areas or time slots;
- Destination-Specific DRT: A service that connects specific destinations such as airports or large office complexes or tourist destinations;
- Substitute DRT: This category of DRT services replaces all or part of the traditional public transport system.

Based on territory orography and demand characteristics, the literature [3,12,26,27] identifies different DRT typologies.

2.2.1. Fixed Route with Bookable Stops

The service illustrated in Figure 1 is characterized by a predetermined route and fixed stops: the minibus will stop exclusively at the stops booked by users who intend to get off or on the vehicle. When the driver does not receive any reservations, the minibus will continue its journey without stopping. This type of DRT service presents the most rigid form and allows drivers to avoid journeys with few or no passengers on board, bringing economic savings to PTAs as well as environmental benefits for the community.

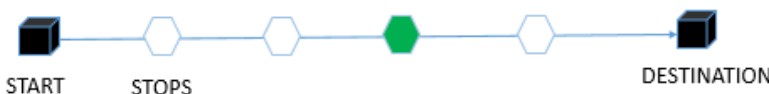

**Figure 1.** Model of fixed route with bookable stops.

2.2.2. Fixed Route with Possible Detours from So-Called Nominal Line

The service in Figure 2 provides a predetermined route from which the driver can deviate to reach any stops distant from the so-called nominal line. This model is particularly suitable for rural contexts where users often request remote stops: this requires the minibus to divert its route from the main road (nominal line) in order to travel on secondary roads and meet the transport demand as a whole.

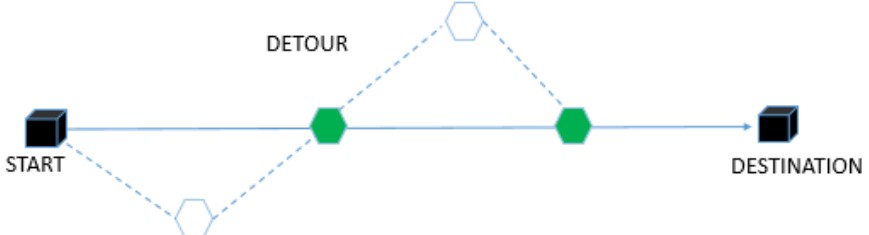

**Figure 2.** Model of fixed route with detours.

### 2.2.3. Variable Route with Fixed Stops

Among the flexible DRT typologies, this model still presents degrees of rigidity: although the route is completely variable, some stops are fixed and predetermined while others are flexible and based on users' requests (Figure 3).

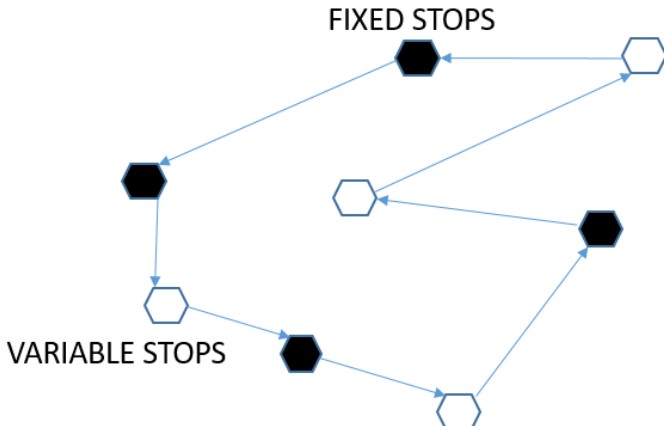

**Figure 3.** Model of variable route with fixed stops.

### 2.2.4. Flexible Models

The most flexible service models that can be introduced with DRT service represent the most expensive solutions for transport companies.

The "One-to-many" model, particularly suitable for commuters who return from the city to their homes during the late afternoon time slot, provides the transport of users from a single predetermined origin to a plurality of destinations (similarly, the "Few-to-many" model carries out the transport from a few origins to a plurality of destinations). This model allows passengers to gather at a single origin (e.g., main square, market, central station, etc.) and be transported to the closest stop to their house (Figure 4).

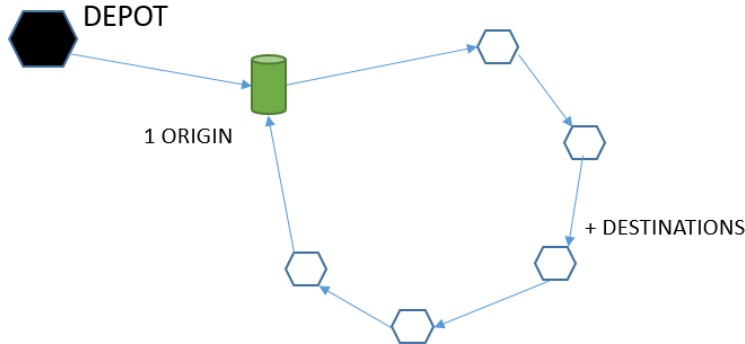

**Figure 4.** "One-to-many" model.

Particularly suitable for the morning time slot are the "Many-to-one" and "Many-to-few" models in which vehicles carry out the passenger transport service from a plurality of

origins to a single common destination, in the first case, or to a few different destinations. These service models find application above all in rural or mountain contexts gravitating around a wider attractive pole, in which the number of commuters who move daily for work/study/health visits is high (Figure 5).

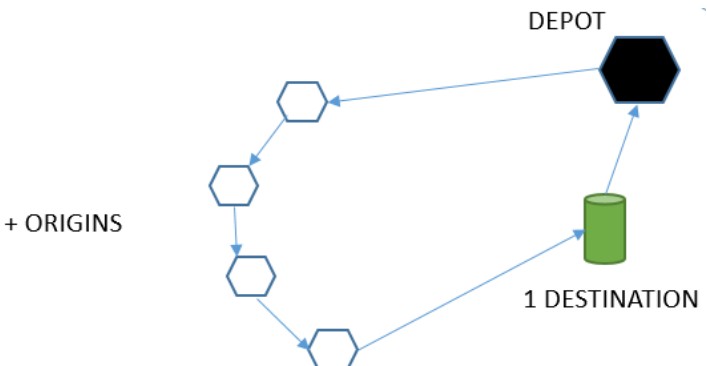

**Figure 5.** "Many-to-one" model.

In conclusion, the most flexible solution available to PTAs to perform DRT service, and particularly used in the transport of people with disabilities due to its flexibility characteristics, is the so-called "Many-to-many" model (Figure 6): drivers carry out the transport service from a plurality of origins to a plurality of destinations by picking up people directly at their home and dropping them off at the final destination (real door-to-door service).

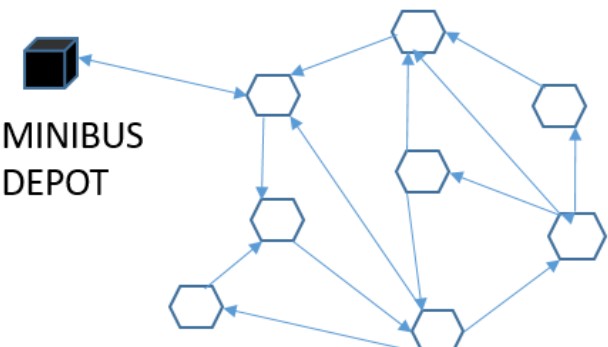

**Figure 6.** "Many-to-many" model.

## 3. Materials and Methods

### 3.1. Performance Indicators Found in the Literature

From an analysis of the scientific literature emerge various Key Performance Indicators (KPIs), which are used to evaluate the performance of on-demand transport services. Table 1 shows the main KPIs identified in the literature: all of the KPIs are related to DRT service carried out by minibuses managed by local PTAs except two case studies examined [28,29], which concern ride-sharing platforms.

**Table 1.** Key Performance Indicators.

| Key Performance Indicator | Author | | | | | | | | | | |
|---|---|---|---|---|---|---|---|---|---|---|---|
| | Alonso-Gonzalez et al. 2018 [14] | Calabrò et al. 2020 [30] | Dias et al. 2012 [31] | Feizi et al. 2022 [32] | FlexDanmark 2013 [33] | Fuchs 2020 [34] | Guan et al. 2018 [28] | Inturri et al. 2019 [35] | Kirsimaa and Suik 2020 [36] | Linares et al. 2017 [29] | Westerlund 2016 [37] |
| Coverage, routing and links to other modes | x | | | | | | | | | x | |
| Operating hours | x | | x | | | x | | | | x | |
| Passenger satisfaction | | x | | | | | | | x | | x |
| Vehicle characteristics | x | | | | | x | | | | | |
| Share of Declined Trips | x | x | | x | | | x | | | x | |
| Total number of transported passengers | | x | x | | | | x | x | x | x | |
| Average vehicle load factor | | x | x | x | | | | x | | | x |
| Total driven distance | | x | | x | | | | x | x | x | |
| Average passenger travelled distance | x | x | x | | | | | x | | | |
| On-time performance | | | | | x | | | | x | | x |
| Average waiting time | | x | x | | | | x | x | | | |
| Average on-board time | | x | x | | | | | x | | x | |
| Average total travel time | x | x | x | | | | | x | | x | |
| Total operating cost | | x | | | | | | x | x | | x |
| Total cost per passenger | | x | | | x | | | x | | | x |

*3.2. KPIs for Our Research*

Based on the literature analysis conducted, the most suitable KPIs were selected for evaluating the strategic nature of DRT service in PTAs' management under observation. Selected KPIs were divided into 3 distinct categories: "DRT system characteristics", in order to better comprehend the state of the art of a given city in terms of using this technology, "Measures of productivity, efficiency and effectiveness" of the system and "Centrality of DRT in the PTA strategy", to understand management's willingness to invest in this service soon.

KPIs related to the first category mentioned above are "Coverage and routing", useful for outlining DRT service lines and service models used, "Operating hours" (number of hours of service performed per day), "Vehicle characteristics" (typology of means of transport), "Booking system" (options available to users to book rides), "Number of DRT users/year", "% usage of DRT compared to Fixed Transit" and "Types of DRT trips made" (typology of DRT model). For the second category mentioned, the following KPIs were selected: "Passenger trips per vehicle-hour" (total passenger trips/total vehicle-hours), "Operating cost per vehicle-hour" (total operating cost/total vehicle-hours) or "Operating cost per vehicle-mile" (total operating cost/total vehicle-miles) and "Operating cost per passenger trips" (total operating cost/total passenger trips). For the third category, three KPIs were selected: "Fleet size of DRT vehicles", to understand in absolute value the number of vehicles dedicated to DRT service; "Contribution to growth "(service DRT growth year/total PTA growth), helpful to comprehend the annual contribution of DRT

service, in terms of users/revenues, to the general growth of the PTA; and finally, KPI "PTA expenditures in DRT per capita" (annual expenditure in DRT/population), in order to compare investments made in DRT service by PTAs with those made by other PTAs in cities with different populations.

## 4. Results

### 4.1. DRT System Characteristics

The KPIs related to the "DRT system characteristics" category were used to compare DRT services in Tampere, Braunschweig and Genoa from a technical and structural point of view. The data were collected by visiting the official websites of PTAs and through direct contact with managers responsible for on-call transport within the three transport companies studied.

#### 4.1.1. DRT in the City of Genoa, Italy

In Genoa, urban and metropolitan public transport is managed by AMT S.p.a. Table 2 shows the number of lines managed by the transport company, the vehicle fleet and the network at its disposal.

**Table 2.** AMT public transport offer.

| Service Lines | Vehicle Fleet | Network |
|---|---|---|
| 277 bus lines | | |
| 8430 stops | | |
| 654 terminus | | |
| 1 metro line | | |
| 2 cable railways | 898 buses in total | |
| 12 lifts | ○ 881 buses | 2503 km in total |
| 1 rack railway | ■ 37 electric | ○ 25.3 km by rail |
| 1 fast line by sea | ■ 11 hybrid | ○ 13.4 km by trolleybus |
| (Navebus) | ○ 17 trolleybuses | ○ 7.2 km by metro |
| 1 railway Genoa-Casella | 25 metro trains | ○ 1.8 km by cable railway |
| 2 airport lines (Volabus | 2 rack railways vehicles | |
| and Flybus) | 4 cable railway vehicles | |
| 31 supplementary services | | |
| 11 areas served by DRT | | |
| service (DrinBus and | | |
| Chiama il Bus) | | |

Source: Own elaboration based on [38].

As can be seen from Table 2, on-demand transport of AMT currently concerns 4 different areas of the city and the use of 8 vehicles (with both 8 and 14 seats). Originally, the first pilot conducted in Genoa was tested in April 2002 and concerned only two areas of the city (Pegli and Quinto), quite distant from the city centre and, at the time, inadequately served with traditional public transport. Based on the success of the first trial, on-call service was then extended to assume the current conformation which covers four different areas of the city (each including various neighbourhoods): to the west Pegli and Multedo and to the east Quarto, Quinto and Nervi. Over time some poorly served areas of Valbisagno and Valpolcevera were also added to the DRT network (Figure 7).

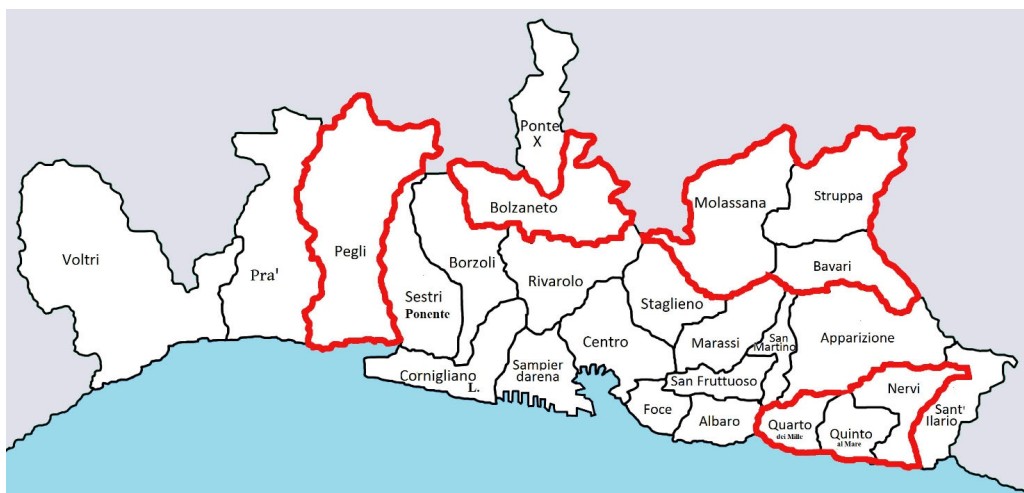

**Figure 7.** DRT service areas in Genoa [39].

DRT service in Genoa is active from h. 7 am to 8 pm every day of the week except Sunday (and holidays) for the eastern and western areas. In Valpolcevera, the service starts one hour in advance (h. 6 a.m.–8 p.m.), while in Valbisagno, it is active only in the evening from h. 9 p.m. to 12 a.m. Users who wish to book their ride can call the toll-free number from Monday to Saturday from h. 6 a.m. to 11.30 p.m. Passengers can book one or more trips for the same day (at least 30 min in advance), for the current week or the following ones. If users have not booked the ride by phone and still show up at the stop, they can board the minibus as long as there are still free seats and they accept the previously planned route. Passengers can also book rides using the "AMT Servizi a chiamata" app, available 24 h a day.

Regarding the cost of the ticket, passengers must bear in addition to the payment of the ordinary ticket for traditional public transport a supplement of EUR 1 per trip (with daily duration) except for Valbisagno, which provides, having only the evening service, a supplement of EUR 0.50. A test is currently underway in Valpolcevera to eliminate the surcharge: users can board the "DrinBus" vehicles paying only the cost of AMT ordinary ticket.

4.1.2. DRT in the City of Tampere, Finland

Public transport in the Tampere region is organized by Nysse-Tampere regional transport. The public transport system includes 72 bus routes, 2 tramlines, 3 commuter train lines and a city bike system with 700 bikes. There are eight DRT systems within the Nysse region, one in each of the Nysse region municipalities. The largest DRT system within the city of Tampere is the focus of this study. The Tampere DRT system PALI includes 18 service areas, shown in Figure 8, as well as 2 fixed route services. PALI is organized by Tuomi Logistiikka, a company owned by the Tampere region municipalities organizing various public purchases in the region.

Most DRT service areas have one or two fixed stops but variable routes and operate on weekdays from 8.30 a.m. to 2.30 p.m. Each service area is operated mainly by one low-floor minibus with 14–16 seats.

Passengers can book the PALI ride one week in advance and preferably on the day before the ride, but also the same day by calling a general service number from 8 a.m. to 5 p.m. on weekdays for nine service areas and directly to the driver on the rest of the service areas. Passengers can also board the minibus at the fixed stop or wave the bus to stop at any place without booking. In case the bus is full, priority is given to passengers who have booked the ride in advance.

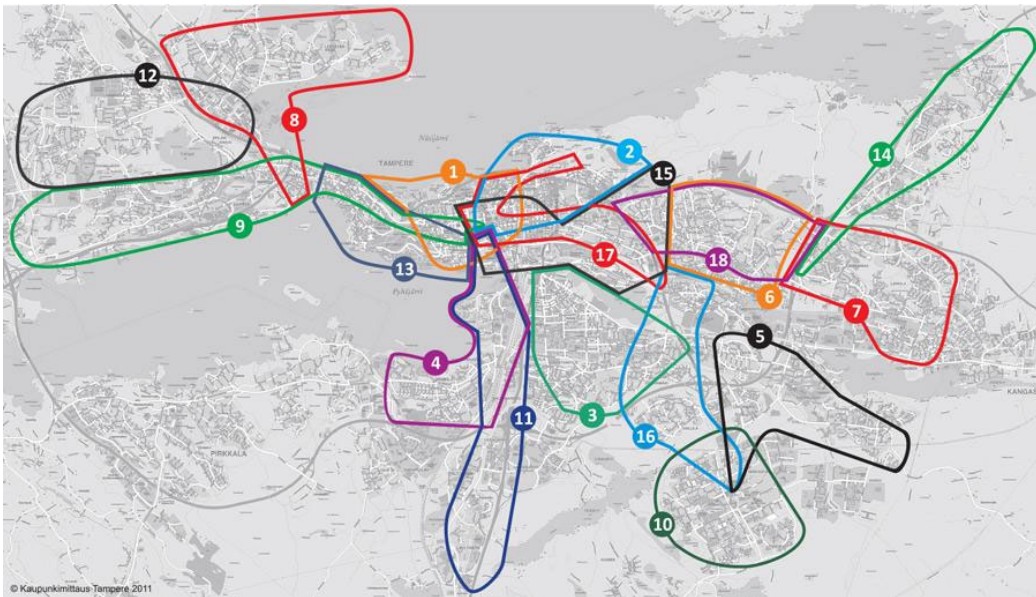

**Figure 8.** DRT service areas in Tampere [40].

The ride costs the same as fixed public transport tickets, i.e., EUR 3.50 for an advance purchase single ticket, EUR 2.10 for using a travel card and EUR 1.16 for seniors (aged 65 or older, between 9 a.m. and 2 p.m. only). Passengers using a wheelchair and their assistants, as well as the assistants of passengers with disabilities, can travel free of charge.

In Tampere city, which has around 240,000 inhabitants, 19% of whom are aged over 64, the DRT system was used for 240,000 trips annually before COVID-19 and the number of trips decreased by 26% in 2020 to 180,000 trips, with a slight increase to 190,000 trips in 2021. This is just 0.6% of the total of 41.2 million trips made in Nysse in 2019 and 27.8 million trips in 2020. Hence, the total number of Nysse trips decreased during COVID-19 by 33%, which is slightly higher than the decrease in the Tampere DRT system PALI.

### 4.1.3. DRT in the City of Braunschweig, Germany

Braunschweig is a city in northern Germany with a population of about 250,000 [41]. Public transport in the Braunschweig region is provided by the Regionalverband Großraum Braunschweig as the transport authority. This is responsible for local rail passenger transport and local public road passenger transport. In addition, various transport companies provide transport services with buses and trams. For further reference, only the Braunschweiger Verkehrs-GmbH (BSVG) and the Kraftverkehrsgesellschaft mbH Braunschweig (KVG) are mentioned here. The BSVG regulates public transport with busses and trams in the inner city and partly in the surrounding area of Braunschweig. The KVG operates urban transport in surrounding centres in the greater Braunschweig area, including lines to and from Braunschweig [42].

In the service area of the BSVG and KVG, dial-a-ride transportation with taxis or small buses operate on various routes (around 40 lines) [43]. The service area of the dial-a-ride transport from BSVG and KVG is in the areas Salzgitter, Helmstedt, Wolfsburg, Wolfenbüttel and Goslar. They operate at off-peak times when it is not profitable to provide a regular service. These operate according to a fixed timetable and route. They must be registered 45 min before the ride by phone or via an app. Ticket prices are based on the tariffs of the Verkehrsverbund Region Braunschweig (VRB). Passengers only need one ticket for a journey within the area of Regionalverband Großraum Braunschweig, regardless of the transport company. The fare is calculated according to a uniform tariff system from the VRB. There is a city tariff (EUR 2.90) for Braunschweig, Wolfsburg and Goslar and four different tariff zones. If the trip takes place within one tariff zone, the passenger needs a ticket for the fare stage 1 (EUR 3). If the trip takes place across several tariff zones, the fare

goes up to EUR 9.70 for an adult. For dial-a-ride transportation, a comfort charge of EUR 1 per trip and passenger is also charged [44].

As of 2021, there is an additional offer for flexible transportation in the greater Braunschweig area. Flexo is an on-demand transport service with barrier-free minibuses. The buses can currently be booked by phone and will be converted to an app in the future. There are no fixed departure times. Instead, the next journey is based on the requests of the passengers. There are also no fixed routes, only fixed stops. Therefore, already existing bus stops as well as new bus stops are used for this service. Flexo also operates in the VRB tariff zone. The bus services vary depending on the line, but on average, each line operates between 5 am and 10 pm during the week. On Saturdays and Sundays, flexo runs later in the morning and stops earlier in the evening [45].

Figure 9 shows the service area of the Greater Braunschweig area as well as the flexo service area. The abbreviation SZ stands for the city of Salzgitter.

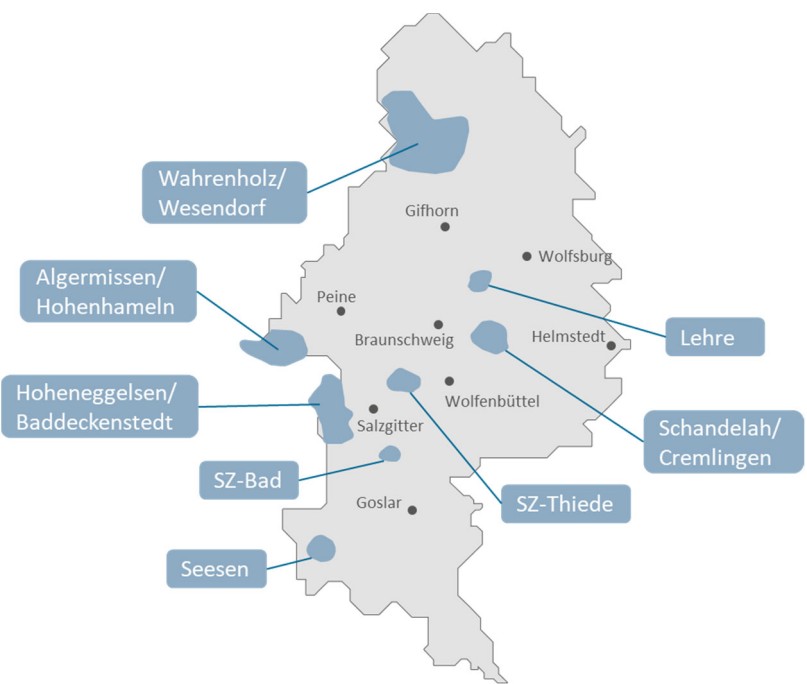

**Figure 9.** Operation area Regionalverband Großraum Braunschweig and flexo service areas [45].

4.1.4. Comparison of System Characteristics

As can be seen above, there are major differences between the cities in DRT system characteristics. In terms of coverage, Tampere's PALI system is more focused in urban areas as a supplementary system for the elderly and the disabled with very limited operating hours and standard public transport fare prices compared to the wider operating hours and additional prices in Genoa. In Braunschweig and the surrounding area, the systems are more used for rural areas as feeders to train stations or bus stops that connect to the cities.

In terms of DRT typology, the Tampere system consists mainly of fixed stops and variable routes, whereas the Genoa system operates mainly on fixed routes and bookable stops. In Braunschweig and the surrounding area, the systems also run according to fixed routes and timetables as required, but also according to bookable stops. Minibuses are used in all cities, but Tampere has a bit larger vehicles, while in Braunschweig, sometimes only taxis operate on the lines. The booking system is centralized in Genoa, while in Tampere, half of the service areas used centralized booking and there are plans to include other service areas in to the centralized system, as well. In Braunschweig, each service is booked differently depending on the transport company. In both Tampere and Genoa, bookings are made via telephone, but also non-booked trips are possible. An app is also considered in Tampere and Braunschweig (flexo) as a booking method in the future, and for the operation

area of the KVG, a passenger can use the app "VRB—Bus & Bahn". In Genoa, an app called "AMT Servizi a Chiamata" was launched in June 2022 as an additional booking possibility.

The number of DRT trips per inhabitant aged over 64 was 5.3 in Tampere, while the share of DRT trips of total public transport trips was 0.6%. In Genoa, the operating hours are 13/day, the number of DRT users/year is 96,200 (with almost 50,000 bookings/year) and compared to FT, the usage of DRT is 0.09% (based on the number of passengers carried). In Braunschweig (KVG only), 6434 journeys were booked with a total of 10,580 passengers and 127,346 km in the year 2021 [46].

*4.2. DRT Performance in Case Areas*

4.2.1. Measures of Productivity, Efficiency and Effectiveness

For the second category of comparison, i.e., measures of productivity, efficiency and effectiveness, the following KPIs were selected: "Passenger trips per vehicle-hour" (total passenger trips/total vehicle-hours), "Operating cost per vehicle-hour" (total operating cost/total vehicle-hours) or "Operating cost per vehicle-mile" (total operating cost/total vehicle-miles) and "Operating cost per passenger trips" (total operating cost/total passenger trips).

In Tampere, the number of trips per vehicle-hour was on average 5.7 in 2021, with wide variation from 1.8 to 11.5 trips per vehicle-hour between the service areas. In Genoa, the passenger trips per vehicle-hour are 8.74 (comparing the total number of passengers transported and the total hours of actual service performed during the year). Operating costs per vehicle-hour are in Tampere on average EUR 41.50/h, which includes the profit of Tuomi Logistiikka, the company purchasing the DRT services for the city of Tampere. In the tendering bids, the costs of DRT services have been between EUR 36.60/h and EUR 37.10/h. In Genoa, operating costs per vehicle hour are EUR 29/h (EUR 32/h if including call centre management and general administrative costs). There are no mileage data available from Tampere and Genoa to calculate the costs per vehicle/kilometre. Operating costs per passenger trip were in Tampere EUR 4.98/trip before COVID-19, but increased to more than EUR 7/trip in 2020–2021, and in Genoa are EUR 3.65/trip (if considering the total operating costs including call centre and administrative expenses; otherwise, EUR 3.31/trip).

In the KVG area, 41 vehicles operate on all lines. Depending on the line, these are differently utilized and in operation. The costs for all vehicles amount to EUR 176,032 and on average approx. EUR 4300 per vehicle. The average cost per kilometre is EUR 1.38, the average cost per order is EUR 27.36 and the average cost per person is EUR 16.64 in 2021. The income for all lines amounts to EUR 17,443 (2021).

4.2.2. Centrality of DRT in the PTA Strategy

For the third category of comparison, the centrality of DRT in the PTA strategy, three KPIs were selected: "Fleet size of DRT vehicles", "Contribution to growth" (service DRT growth year/total PTA growth) and "PTA expenditures in DRT per capita" (annual expenditure in DRT/population).

In Tampere, the DRT system is operated by 22 minibuses, which is around 10% of the entire Nysse region bus fleet, while in Genoa, it is operated by 8 minibuses (just 0.89% of the whole AMT bus fleet). There are 41 vehicles (minibuses and taxis) in the KVG region and at least one minibus on each flexo line.

In terms of contribution to growth, in Tampere, the number of DRT trips was stable before COVID-19, but the number of trips in fixed public transport grew by 12% in 2016–2019. On the other hand, the number of trips decreased slightly less in DRT than in fixed public transport due to COVID-19. The public transport expenditures in the Nysse region have been around EUR 72 million annually and the share of DRT services is EUR 1.2 million (1.7%). The expenditures into the DRT system in Tampere are EUR 5.10 per inhabitant and EUR 26.70 per inhabitant over 64 years old. Comparative data were unfortunately not available from Genoa and Braunschweig.

In Tampere, the future role of the DRT system is a subject of discussion, because the Finnish regional governance system has been renewed from January 2023 onwards as new regional governments for welfare services begin their operation. The DRT systems may in the future become a part of municipalities' responsibilities, as they have been in the past, or a part of the regional welfare services. In the first case, DRT may be seen as a part of the public transport system and in the latter case a part of the social services. This fundamental decision may lead to very different DRT systems in the future. The current DRT system in Tampere can be seen to have mostly had the role of a social service rather than public transport service. Although anyone can use the PALI minibuses, the marketing is mostly directed toward the elderly and the disabled.

Another interesting future development discussed in the interviews relates to vehicles. Electric buses are strongly pushed within the European Union by the Clean Vehicles Directive and its mandatory share of battery electric vehicles. Furthermore, the recent increase in diesel prices drives operators to seek to invest in battery electric buses. However, there seems to be a shortage of suitable electric minibuses with 14–16 seats, although there are plenty of electric minibuses with eight seats and electric full-sized buses.

In Genoa, DRT is taking on an increasingly central role for the local public transport company. As evidence of this, it should be noted that on 1 January 2021, AMT S.p.a. acquired the provincial public transport company ATP (Azienda Trasporti Provinciali) Esercizio S.r.l., which carried out regional transport in the rural areas around the city of Genoa. AMT S.p.a. has thus inherited the DRT services that ATP S.r.l. was starting to implement in some remote areas by deciding to confirm the project and believe in the strategic nature of on-call transport (the first experimental service in Val Graveglia has been active since 14 February 2022). At the urban level, however, the experimentation of the new on-call service dedicated exclusively to the elderly (over 65 years) and defined as "Silver Bus" started on 20 July 2022: the trial is being carried out in two specifically chosen districts of Genoa (Marassi and San Fruttuoso) due to the high density of elderly population and proximity to the Galliera hospital. Despite the success achieved, this service was suspended at the end of the trial period (31 December 2022) due to running out of funds. Amt S.p.a. has plans to extend this service dedicated to the elderly to the entire city area in the coming years.

DRT is becoming more and more important in Germany, especially in the Braunschweig region, especially for rural areas and off-peak times when it is not economically profitable to operate a regular bus service. In the Braunschweig region, a new flexible form of service has been integrated into public transport with the implementation of flexo. It will take some time until potential passengers get used to the new services and prefer to use them. Financing is therefore of fundamental importance in order to secure and operate these services in the long term. This will ensure that they remain an important component and addition to the existing public transport system in the future, especially in rural areas, but also to extend the service time and route of existing lines. In the future, flexo should be extended to other service areas. A major challenge is to acquire sufficient driving personnel, which is also reflected in the current nationwide trend.

## 5. Discussion

This study aimed at finding out if DRT is considered by PTAs as a strategic development area for the future or as an auxiliary service unworthy of investment. Our analysis was based on the DRT key performance indicators found in the literature, and the most important results regarding the three key categories of comparison can be summarized as follows:

- DRT system characteristics vary considerably between the cities of Tampere, Braunschweig and Genoa. Tampere offers good coverage and flexible service within the urban area but with limited operating hours, while in Braunschweig, the focus is on rural areas with more fixed lines and wider operating hours. The PTA of Genoa is the one that invests the most in this technology, having a DRT service valid at

urban, metropolitan and rural levels. The cost for users is the same as for fixed public transport users in Tampere, while in Genoa and Braunschweig, an additional fee of EUR 1 is added. Booking options are flexible in all cities.

- Regarding measures of productivity, efficiency and effectiveness, Genoa shows the lowest cost per passenger with the highest average vehicle utilization and lowest hourly vehicle costs. The values for Tampere are comparable to Genoa, but the costs per trip in Braunschweig seem very high in comparison.
- Regarding the role of DRT in the PTA strategy, there are clear differences. In Tampere, the role and governance of DRT are under consideration due to changes in the social welfare system, but even before these changes, the PTA was not in charge of the DRT service. In Genoa, the PTA is taking more responsibility for the DRT service, and both in Genoa and Braunschweig, new service models are implemented.

Our study has shown that the DRT systems can serve varying roles within public transport services. There seems to be a bit of ambiguity regarding the role of DRT depending on the history and culture of the countries in our comparison. A wider study covering more countries would be needed to gain a deeper understanding of the best practices to ensure efficient and high-quality DRT services in the future.

Looking ahead, the (price) development of the "Deutschlandticket" remains particularly exciting: in May 2023, it is to be a permanent offer following the success of the nine-euro ticket in summer 2022. It is valid throughout Germany for buses and trains in local and regional transport, is part of the third relief package of the German government and is intended to provide financial relief. On the other hand, it is intended to significantly increase the attractiveness of local public transport and provide a greater incentive to switch from car to bus and rail—and thus help achieve the climate targets. DRT options and possible comfort charges remain to be seen.

**Author Contributions:** Conceptualization, T.P., H.L., N.S. and J.P.H.; methodology, T.P., H.L., N.S. and J.P.H.; data curation, T.P., H.L., N.S. and J.P.H.; writing—original draft preparation, T.P., H.L., N.S. and J.P.H.; writing—review and editing, T.P., H.L., N.S. and J.P.H.; supervision, H.L. All authors have read and agreed to the published version of the manuscript.

**Funding:** This research received no external funding.

**Institutional Review Board Statement:** Not applicable.

**Informed Consent Statement:** Not applicable.

**Data Availability Statement:** Not applicable.

**Conflicts of Interest:** The authors declare no conflict of interest.

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
