# Peer review of "The Role of DRT in European Urban Public Transport Systems—A Comparison between Tampere, Braunschweig and Genoa"

_futuretransp, doi:10.3390/futuretransp3020034_

Round 1

Reviewer 1 Report

The paper presents an overview of DRT systems. The paper is structured as a review paper, and it also provides a more in-depth assessment of three DRT systems. A comparative analysis is provided across the services offered in these three contexts.

The paper is well-organised and easy to read. The objectives of the paper are perfectly presented and clearly demonstrated. 

Although I liked reading the draft, I did not find its scientific merit and innovations sufficient as a journal article. As a technical report or possibly a conference paper to locals who have a better sense of the presented DRT systems, I am pretty sure. The paper as it stands can be improved in two ways:

1- Extend the lit review side of the paper and transform it into a full review paper with an international view

2- Extend the comparative analysis to a more extensive set of cities with a meta-analysis on why each city adopted such features and how they can evolve given changes in the conditions. A historical timeline of the evolution of the BRT systems in each city would also be informative. 

Reviewer 2 Report

the manuscript has several grammatical errors and typos 

It is advisable to enter all acronyms in full form when they are mentioned for the first time in the text by inserting the reference acronym (DRT or KPI) next to it.

It is necessary to better emphasise the novelty of the research in the introductory part, better justifying the choice of case studies 

In order to better consider the potential of DRT systems, it is advisable to emphasise their use in areas with weak transport demand and/or as complementary services to public transport, also mentioning what happened in the pandemic phase.

We therefore recommend reading the following works 

1) Campisi, T., Canale, A., Ticali, D., & Tesoriere, G. (2021, March). Innovative solutions for sustainable mobility in areas of weak demand. Some factors influencing the implementation of the DRT system in Enna (Italy). In AIP Conference Proceedings (Vol. 2343, No. 1, p. 090005). AIP Publishing LLC.

2)Campisi, T., Cocuzza, E., Ignaccolo, M., Inturri, G., Tesoriere, G., & Canale, A. (2023). Detailing DRT users in Europe over the last twenty years: a literature overview. Transportation Research Procedia, 69, 727-734.

Sources of the maps used are recommended

We recommend standardising the template of all tables in the text

Round 2

Reviewer 1 Report

They addressed my suggestions

Author Response

Dear Reviewer, thank you for your review.

Reviewer 2 Report

The manuscript still has grammatical errors and typos. once this is corrected, the paper will be eligible for publication 

Author Response

Dear Reviewer we have corrected the text grammar and typos as requested.